# CHALLENGING IMAGES FOR MINDS AND MACHINES

**Amir Rosenfeld, John K. Tsotsos**
Department of Electrical Engineering and Computer Science
York University, Toronto, ON, Canada
`amir@eecs.yorku.ca,tsotsos@cse.yorku.ca`

## ABSTRACT

There is no denying the tremendous leap in the performance of machine learning methods in the past half-decade. Some might even say that specific sub-fields in pattern recognition, such as machine-vision, are as good as solved, reaching human and super-human levels. Arguably, lack of training data and computation power are all that stand between us and solving the remaining ones. In this position paper we underline cases in vision which are challenging to machines and even to human observers. This is to show limitations of contemporary models that are hard to ameliorate by following the current trend to increase training data, network capacity or computational power. Moreover, we claim that attempting to do so is in principle a suboptimal approach. We provide a taster of such examples in hope to encourage and challenge the machine learning community to develop new directions to solve the said difficulties.

## 1 INTRODUCTION

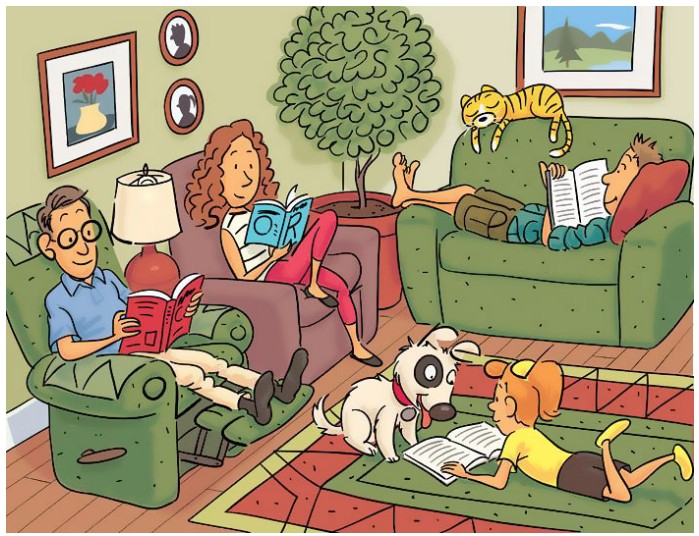

Figure 1: A children's puzzle where the goal is to find six hidden words: Book, words, story, pages, read, novel. For a machine this is far from child's play. Could this be solved by providing a million similar examples to a deep-learning system? Does a human need such training?

Once only known to a few outside of academia, machine-learning has become ubiquitous in both popular media and in the industry. Superhuman capabilities are now being gradually recorded in various fields: in the game of GO, (Silver et al. (2016; 2017)), in face verification (Lu & Tang (2015); Qi & Zhang (2018)), image categorization (He et al. (2015)) and even in logical reasoning in simple scenes (Santoro et al. (2017); Perez et al. (2017a;b)).

Most current leading methods involve some variant of deep learning. Consequentially, they require large amounts of hand-labeled data (with the exception of Silver et al. (2017) - which used

Figure 2: Variants of textual CAPTCHA. Captchas are becoming increasingly difficult (reproduced from Le et al. (2017))

self-play to gain experience). This has elicited a data-hungry era, with increasingly large-scale datasets painstakingly labeled for object classification/detection/segmentation, image annotation, visual question-answering, and pose estimation (Russakovsky et al. (2015); Lin et al. (2014); Krishna et al. (2017); Antol et al. (2015); Güler et al. (2018)) to name a few. This is accompanied by a growing demand for computational power.

We bring forward challenges in vision which do not seem to be solved by current methods - and more importantly - by current popular methodologies, meaning that neither additional data, nor added computational power will be the drivers of the solution.

RELATED WORK

**Imbalanced or Small Data:** datasets tend to be naturally imbalanced, and there is a long history of suggested remedies (Lim et al. (2011); Zhu et al. (2014); Wang et al. (2017)). Handling lack of training data has also been treated by attempting to use web-scale data of lesser quality than hand-annotated dataset Sun et al. (2017), simulating data [cite data for cars, text recognition in the wild, captcha]. **Transfer Learning:** reusing features of networks trained on large is a useful starting point (cf Sharif Razavian et al. (2014)) **One-Shot-Learning**: attempting to reduce the number of required training example, in extreme cases to one or even zero examples (Snell et al. (2017)); **Deep-Learning Failures**: recently, some simple cases where deep learning fails to work as one would possibly expect were introduced, along with theoretical justifications (Shalev-Shwartz et al. (2017)).

## 2 CHALLENGING CASES

We present two examples and then discuss them. They have a few common characteristics: humans are able to solve them on the first "encounter" - despite not having seen any such images before. Incidentally - but not critically - the two examples are from the domain of visual text recognition. Moreover, though humans know how to recognize text as seen in regular textbooks, street-signs, etc, the text in these images is either hidden, rendered, or distorted in an uncharacteristic manner.

**Children's games**: the first case is well exemplified by a child's game, hidden word puzzles. The goal is to find hidden words in an image. Fig. 1 shows an arbitrarily selected example. For a human observer this is a solvable puzzle, though it may take a few minutes to complete. We applied two state-of-the-art methods for text recognition in the wild with available code (Shi et al. (2017)) or an on line-demo (Zhou et al. (2017)[1]) on the image in Fig. 1. As this did not work immediately, we focused on the word "NOVEL" (the "N" is below the forearm of the left person, ending with an "L" below his foot), by cropping it an rotating so the text is level, cropping more tightly , and even cropping only the letter "L". See Table 1 for the corresponding sub-images (including the entire image at the top row) and the results output by the two methods.

This is by no means a systematic test and some may even claim that it isn't fair - and they would be right: these systems were not trained on such images; Shi et al. (2017) was only trained on a photo-realistic dataset of 8 million synthetic training images, and Zhou et al. (2017) was only trained on tens of thousands of images from coco-text (Veit et al. (2016)), or used powerful pre-trained networks where training data was less available.

**CAPTCHA**: a well-known mechanism to thwart automated misuse of websites by distinguishing between humans and machines (Von Ahn et al. (2003)). Textual captchas involve presenting an image of text which has to be read and written by the user. We focus on this type of captcha, though others exist (Singh & Pal (2014)). The introduction of captchas immediately triggered the invention of new automatic ways to break them (Mori & Malik (2003)), which eventually sparked an "arms

---

[1] http://east.zxytim.com

| Sub Image | 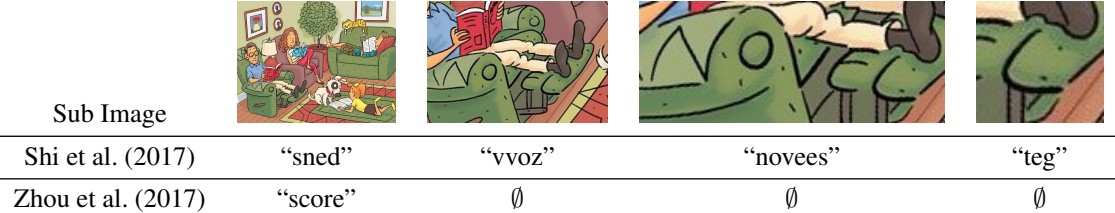 | | | |
|---|---|---|---|---|
| Shi et al. (2017) | "sned" | "vvoz" | "novees" | "teg" |
| Zhou et al. (2017) | "score" | ∅ | ∅ | ∅ |

Table 1: Text detected by two state-of-the-art scene-text recognition methods applied to sub-images of a children's puzzle. ∅ means no text was detected by the method (images scaled to fit figure).

race" between increasingly complex captchas and correspondingly powerful automated methods (Chen et al. (2017)). This caused a state where on one-hand the best leading textual captcha-solution methods involve training DNN's over data with similar distortion characteristics as the desired types of captcha - though still these systems have limited success rates (at times less than 50%) - and on the other hand the level of distortion has become such that humans have a hard-time solving some of them.

## 3 MACHINES VS HUMANS AS SUPERVISED LEARNERS

One can rule out the suggested examples by saying that they are simply out-of-sample datapoints on behalf of a statistical learner's perspective. Yet it seems that with whatever supervision human-beings receive - they are usually able to solve them despite not being especially exposed to this kind of stimulus. Moreover, precisely these kinds of images are used routinely in human IQ testing, so they are a universally accepted indicator for human performance. If these examples may seem esoteric, we can revert to more common cases: as a child, how often is one exposed to bounding boxes of objects? How often to delineations of objects with precise segmentation masks? How often to pose-configurations, facial and bodily key-points, and dense-meshes of 3D objects overlayed on their field of view (Güler et al. (2018))? More critically, for how many different object types does this happen (if any), for how many different instances, with what level of precision of annotation, and in how many modalities?

The granularity of visual supervision given to machines seems to be much finer than that given to humans. As for the amount of directly supervised data, it does not seem to really be the main limiting factor; as already noted several times, performance either saturates with training data (Zhu et al. (2012; 2016)) or at best grows logarithmically (Sun et al. (2017); Hestness et al. (2017), increasing mAP from 53% to 58% when growing from 10M to 300M examples) making the solution of more data for better performance simply impractical - even for those with the most resources. And this is for "common" problems, such as object detection.

Humans who only ever read street-signs and textbooks are able to solve captchas of various kinds without any special training on their first encounter with them. The same is true for the "picture puzzles" mentioned above, as it is for other cases not mentioned here. We do not claim that humans are not subject to supervised learning in their early life, and in later stages. On the contrary, super-visory signals arise from multiple sources: caretakers who provide supervisory signals by teaching, "internal supervision" provided by innate biases (Ullman et al. (2012)) and finally rewards stemming from results of behaviour, such as suffering pain from hitting an object. But any such supervision is interspersed within a vast, continuous stream of unsupervised data, most of which does not have an easily measurable supervisory affect on the observer.

There is something fundamentally different about the way humans construct or use internal representations, enabling them to reason about and solve new pattern-recognition tasks. We hypothesize that these are approached by generating procedures of a compositional nature when presented with a novel - or known - task (as suggested by the Visual Routines of Ullman (1984) or the Cognitive Programs of Tsotsos & Kruijne (2014). We intend to maintain a collection of examples beyond the ones suggested above, to encourage the community to attempt to solve them, not by learning from vast amounts of similar examples, but by learning from related, simpler subtasks and learning to reason and solve them by composing the appropriate solutions.

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
