# OpenReview forum: "Challenging Images For Minds and Machines"
_ICLR.cc/2018/Workshop — Reject_

### Official Review · AnonReviewer1 · 2018-02-25
**A position paper about the**

**Rating:** 4
**Confidence:** 5

**Review:**

This position paper highlight limitations of current deep approaches to learn representations that can transfer well to new test sets. To cite : "There is something fundamentally different about the way humans construct or use internal representations, enabling them to reason about and solve new pattern-recognition tasks." The authors provide visual text recognition examples that are very hard for machines and people.

I agree with the points made by the authors. I also believe that most authors in the ICLR community are aware of these (and other) limitations. There are continuous discussions about these topics, also in the context of life-long-learning, trasnfer learning, and general-AI. The challenge is to invent the right algorithms, tasks and datasets that can lead to better transfer.
I feel that more technical content should be provided for such a paper to be interesting to the ICLR audience.

---

### Official Review · AnonReviewer2 · 2018-03-10
**A paper around a single image**

**Rating:** 4
**Confidence:** 5

**Review:**

This paper aims to present tasks that are difficult for current computer vision systems to solve. The authors argue that these tasks cannot be solved by adding more supervision/data.

Pros
- The example provided by the authors (Figure 1) is interesting. The OCR methods perform very poorly in the wild and the authors do a good job of highlighting this (Table 1)

Cons
- The observation that adding more supervision will not solve machine vision is a popular view in the community, and has been the case for a while. This has been a fairly consistent message across the years, so this paper pointing this out is not surprising. Please also refer to https://arxiv.org/abs/1604.00289 which nicely summarizes this. It's also a paper worth citing.

- The entire paper presents only one "challenge" - Figure 1 (a single image). Essentially, that is the only image the authors talk about and the only concrete "problem" they talk about. The title of the paper has the word "images", which again is not true.

I would encourage the authors to highlight problems (themes) which will be valuable for the community.

---

### Decision · Program_Chairs · 2018-03-20
**ICLR 2018 Workshop Acceptance Decision**

**Decision:**

Reject

**Comment:**

Based on the reviews, this paper has not been accepted for presentation at the ICLR workshop. However, the conversation and updates can continue to appear here on OpenReview.